# Injecting Domain Knowledge from Empirical Interatomic Potentials to Neural Networks for Predicting Material Properties

**Zeren Shui**
University of Minnesota
shuix007@umn.edu

**Daniel S. Karls**[*]
University of Minnesota
karl0100@umn.edu

**Mingjian Wen**[*]
University of Houston
mjwen@uh.edu

**Ilia A. Nikiforov**
University of Minnesota
nikif002@umn.edu

**Ellad B. Tadmor**
University of Minnesota
tadmor@umn.edu

**George Karypis**
University of Minnesota / AWS
karypis@umn.edu

## Abstract

For decades, atomistic modeling has played a crucial role in predicting the behavior of materials in numerous fields ranging from nanotechnology to drug discovery. The most accurate methods in this domain are rooted in first-principles quantum mechanical calculations such as density functional theory (DFT). Because these methods have remained computationally prohibitive, practitioners have traditionally focused on defining physically motivated closed-form expressions known as empirical interatomic potentials (EIPs) that approximately model the interactions between atoms in materials. In recent years, neural network (NN)-based potentials trained on quantum mechanical (DFT-labeled) data have emerged as a more accurate alternative to conventional EIPs. However, the generalizability of these models relies heavily on the amount of labeled training data, which is often still insufficient to generate models suitable for general-purpose applications. In this paper, we propose two generic strategies that take advantage of unlabeled training instances to inject domain knowledge from conventional EIPs to NNs in order to increase their generalizability. The first strategy, based on weakly supervised learning, trains an auxiliary classifier on EIPs and selects the best-performing EIP to generate energies to supplement the ground-truth DFT energies in training the NN. The second strategy, based on transfer learning, first pretrains the NN on a large set of easily obtainable EIP energies, and then fine-tunes it on ground-truth DFT energies. Experimental results on three benchmark datasets demonstrate that the first strategy improves baseline NN performance by 5% to 51% while the second improves baseline performance by up to 55%. Combining them further boosts performance.

## 1 Introduction

Predictive modeling of materials is a field with manifold applications that has been the subject of many cross-disciplinary studies. While modeling is conducted at different length scales, all material behavior ultimately has its origins at the nano scale, where the interactions between individual atoms must be understood. The most accurate methods at this scale are based on quantum mechanics theory, requiring explicit consideration of the electronic degrees of freedom described by the Schrödinger equation. However, these methods are presently limited to systems containing at most several

---

[*]Equal contribution

36th Conference on Neural Information Processing Systems (NeurIPS 2022).

thousand atoms, precluding their use in investigation of important microstructural phenomena such as crack propagation. To overcome this limitation, practitioners have long relied upon heuristic models known as *empirical interatomic potentials* (EIPs), which consist of physically motivated analytical functional forms that strive to model the complex electronic interactions between atoms using only the nuclear coordinates of the atoms and their elemental species as input [34]. In the past several years, there has been a surge of interest in the development of machine learning EIPs, particularly those based on neural networks (NNs), as a more accurate alternative to traditional EIPs [4]. In contrast to traditional EIPs, NN potentials contain little inductive bias and, accordingly, require large volumes of training data labeled using first-principles quantum mechanical methods. The first-principles method most commonly used is density functional theory (DFT), which scales as $\mathcal{O}(n_\mathrm{e}^3)$ with the number of valence electrons $n_\mathrm{e}$ in a given configuration of atoms [23]. As a result of this high computational cost, it is difficult to acquire a sufficient number of labeled training instances to create an NN potential that performs accurately over a wide range of applications.

One potential solution to this problem is to seek additional supervision signals. Traditional EIPs are attractive sources of such additional supervision for two reasons. First, their functional forms incorporate prior physical information that allows them to correlate with DFT, and, in regions of relevance to common applications, are often quite accurate. That is, they contain domain knowledge that could benefit the training of NN potentials. Second, EIPs scale linearly with the number of atoms and are thus orders of magnitude faster than DFT, permitting the labeling of massive datasets. Despite the advantages, no research has focused on using EIP supervision signals in training NN potentials.

In this paper, we leverage physically motivated EIPs and unlabeled configurations to tackle the label scarcity challenge for training NN potentials.We propose two generic strategies, weakly supervised learning and transfer learning, for exploiting this additional source of information. In the first strategy, we expand the DFT-labeled training set with unlabeled configurations and their EIP energies. To achieve this goal, we train an auxiliary classifier on the original DFT-labeled training configurations that predicts which one of a selected set of EIPs is likely to produce the most accurate estimate of the DFT energy for each of a large set of unlabeled configurations. The unlabeled configurations are then labeled by their corresponding predicted best-performing EIPs and appended to the training set. We train NN potentials on the expanded training set by optimizing a robust regression loss to mitigate the influence of noise and outliers introduced by the EIP energies. In the second strategy, we adopt a transfer learning approach by way of multi-task pretraining. We first pretrain the representation module of an NN potential to reproduce the energies predicted by the EIPs. During the subsequent fine-tuning stage, the representation module of the NN is paired with a prediction head and trained on the DFT-labeled configurations. These two strategies can be flexibly used and coupled to train any NN potential.

The contributions of this work are three-fold: 1) We demonstrate that EIPs are capable of providing high quality supervision signals for training NN potentials, which opens a new direction for future development of NN potentials; 2) We propose two effective and generic strategies that take advantage of EIPs and unlabeled configurations to tackle the label scarcity challenge for training NN potentials; 3) We conduct comprehensive experiments on three benchmark datasets and four representative NN potentials that cover most of the NN potential forms currently in use. Experimental results show that the proposed strategies successfully inject domain knowledge from EIPs to NN potentials and improve the performance of the NN potentials by up to 55%.

## 2 Preliminaries

### 2.1 Atomic Configurations

The fundamental input in atomistic modeling is an *atomic configuration*. An atomic configuration is a spatial arrangement of atoms $C = \{(Z_i, \mathbf{r}_i)\}_{i=1}^N$ where $Z_i$ and $\mathbf{r}_i$ are the atomic number and the three-dimensional Euclidean coordinates of atom $i$, respectively. In the simplest scenario, an atomic configuration corresponds to an isolated cluster of atoms that comprise a molecule. However, it may more generally describe a bulk system such as a crystal, which contains an infinite number of atoms distributed over space. These systems are modeled using a small collection of atoms in a finite simulation cell that is effectively repeated across all of space with the aid of periodic boundary conditions (PBCs) [34] (see Appendix A.1).

## 2.2 Physics-based Potentials

Mathematically, an EIP is a function, $E = \mathcal{V}(C; \theta)$, that takes an atomic configuration $C$ as input and returns its total potential energy $E$; here, $\theta$ denotes a set of fitting parameters to be determined. The functional form $\mathcal{V}$ of a physics-based EIP is made up of carefully designed analytic expressions that strive to capture the underlying physics in the material it models [43, 44]. Because the functional form itself is intended to capture most of the relevant physics, such models need relatively few parameters (typically on the order of ten) and are usually fitted to a set of material properties deemed most relevant to real-world applications. It is expected that physics-based EIPs will approximate the first-principles energy surface well in the vicinity of atomic configurations corresponding to the material properties to which they were fit. However, their generalizability can be inconsistent, as shown in Fig. 1.

## 2.3 Machine Learning Potentials

In contrast to physics-based EIPs, machine learning EIPs employ general-purpose regression algorithms as the functional form $\mathcal{V}$ that do not encode any knowledge of the material it models. Therefore, machine learning EIPs are almost exclusively trained on large sets of DFT data so as to include as much physical knowledge as possible.

One important class of machine learning EIPs are neural network (NN)-based potentials. In this paper, we define an NN potential $\mathcal{M} : \mathcal{C} \mapsto \mathcal{E}$ as

$$\mathcal{M}(C) = f_{\text{pred}} \circ f_{\text{rep}}(C),$$

where $f_{\text{rep}} : \mathcal{C} \mapsto \mathcal{R}^{n \times d}$ is the representation learning module that maps each of the $n$ atoms of $C$ to a feature vector of length $d$ based on its local environment, and $f_{\text{pred}} : \mathcal{R}^{n \times d} \mapsto \mathcal{E}$ is the prediction module that maps the feature vectors of the atoms to the total potential energy.

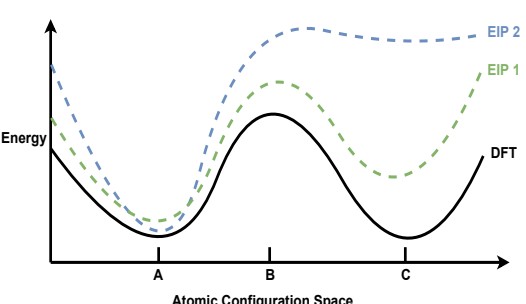

Figure 1: Schematic illustration of the energy landscape defined over atomic configuration space by two physics-based EIPs and by DFT. Both EIPs are fitted to reproduce properties of the DFT energy landscape near atomic configuration A. Away from configuration A, the relative accuracy of the EIPs compared to DFT varies: at point B, EIP 1 is a fair approximation while EIP 2 is less accurate; at point C, neither EIP is accurate.

A concrete instance of this type of model are those that use a graph neural network (GNN) [24, 16, 40, 31, 21] as the representation learning module and a multilayer perceptron (MLP) as the prediction module. Before being passed to the representation learning module $f_{\text{rep}}$, an atomic configuration $C$ is first converted to a graph $\mathcal{G} = (V, E)$ with nodes $V$ and edges $E$. One node is defined for each atom in the simulation cell, as well as for additional padding atoms representing PBCs if present. An edge is created between any two nodes with a distance smaller than a prescribed cutoff radius. Next, for each atom $i$, its atomic number $Z_i$ is one-hot encoded into an initial feature vector $h_i^{(0)}$ of the node corresponding to the atom. The GNN representation learning module $f_{\text{rep}}$ then updates the feature vectors using a message passing paradigm [15], i.e., the nodes iteratively aggregate information from their neighbors. Formally, the feature vector of node $i$ at the $l + 1$-th layer $\mathbf{h}_i^{(l+1)}$ is updated as a function of the feature vectors of its neighbors $\mathcal{N}(i)$ and itself at the previous layer,

$$\mathbf{m}_i^{(l)} = \text{Agg}^{(l)} \left( \left\{ f^{(l)} \left( \mathbf{h}_i^{(l)}, \mathbf{h}_j^{(l)}, \mathbf{e}_{ij}^{(l)} \right) \mid j \in \mathcal{N}(i) \right\} \right), \tag{1}$$

$$\mathbf{h}_i^{(l+1)} = \text{Update}^{(l)} \left( \mathbf{h}_i^{(l)}, \mathbf{m}_i^{(l)} \right), \tag{2}$$

where $f^{(l)}$ and $\text{Update}^{(l)}$ are learnable functions, $\text{Agg}^{(l)}$ is a permutation-invariant function that operates on sets of feature vectors, and $\mathbf{e}_{ij}^{(l)}$ denotes the feature vector associated with the edge connecting node $i$ and node $j$ at the $l$-th layer. Finally, the prediction module $f_{\text{pred}}$ maps the feature vector of each atom at the last layer $\mathbf{h}_i$ to a corresponding energy contribution and sums them to arrive at the total energy of the configuration, i.e., $E = \sum_i \text{MLP}(\mathbf{h}_i)$.

# 3 Related Works

## 3.1 Neural Network Potentials

The first modern NN potential was proposed by Behler and Parrinello [3]. In their formulation, an atomic descriptor (i.e., basis functions that transform an atomic configuration into a fixed-length fingerprint vector) based on the bond lengths and bond angles is passed to an MLP. On top of this formulation, a Monte Carlo dropout technique can be applied to the MLP to equip the potential with the ability to quantify its predictive uncertainty [42]. The DeePMD method is a similar approach to that of Behler and Parrinello, except that a novel atomic environment descriptor is used [48]. More recently, researchers have developed GNN potentials based on a message passing paradigm [30, 46]. Another class of GNN potentials such as NequIP [2] and GemNet [14] pass equivariant messages rather than invariant ones based on the formulation of tensor field networks [37] and achieve state-of-the-art performance. All of these models are trained with supervised learning, without exploring the possibilities of leveraging weakly supervised learning or transfer learning to take advantage of unlabeled data.

## 3.2 Weakly Supervised Learning

Weakly supervised learning refers to techniques that attempt to train machine learning models from incomplete (only a portion of training instances are labeled) or inaccurate (noisy labels) supervision signals [50]. Solutions for incomplete supervision usually fall into the category of semi-supervised learning, which assumes that nearby instances have similar labels [25, 33, 29]. For inaccurate supervision, a model either learns directly from noisy labels with noise-robust algorithms [12, 49, 13] or resorts to a small portion of clean labeled data to reduce the noise [39, 45].

## 3.3 Transfer Learning

Transfer learning [51] refers to a machine learning paradigm that transfers knowledge a model learns from one or more relevant tasks to benefit a target task. Transfer learning has enjoyed great success, especially in the low-data regime, as demonstrated by the rise of pretrained neural networks. This recent trend began with natural language processing when BERT [9] and successive large pretrained language models [26, 7] were released and quickly gained popularity in other domains such as computer vision [17, 8, 10] and graph learning [47, 19, 28]. These methods pretrain large neural networks on self-supervised tasks in order to encode common contextual knowledge in the structured input. Another approach imparts domain-specific knowledge by pretraining models on tasks related to the target task but for which abundant labeled data is available [18, 27]. The pretrained model is then fine-tuned on the limited training data of the target task.

# 4 Methods

## 4.1 Problem Definition

Let $\mathcal{P}$ be a set of physics-based EIPs, $\mathcal{C}$ be the space of all possible atomic configurations, $\mathcal{C}_{\text{DFT}} = \{C_i, \{E_i^p\}_{p \in \mathcal{P}}, E_i^{\text{DFT}}\}_{i=1}^m$ be a set of $m$ configurations with corresponding DFT and EIP energies, and $\mathcal{C}_{\text{EIP}} = \{C_i, \{E_i^p\}_{p \in \mathcal{P}}\}_{i=m+1}^{m+n}$ be a set of $n$ configurations with only physics-based EIP energies. Here, $E_i^{\text{DFT}}$ and $E_i^p$ denote the energies predicted for configuration $C_i$ by DFT and the $p$-th physics-based EIP, respectively. In practice, physics-based EIPs are much less expensive than DFT, and so the size of $\mathcal{C}_{\text{EIP}}$ is much larger than $\mathcal{C}_{\text{DFT}}$, i.e., $n \gg m$. Our goal is to use this data from physics-based EIPs to train an NN-based EIP $\mathcal{M}$ to closely approximate the DFT energy surface over the space of all atomic configurations, i.e.,

$$\mathcal{M}(C) \approx E^{\text{DFT}}, \forall C \in \mathcal{C}.$$

## 4.2 Label Augmentation

Because physics-based EIPs are developed as approximations to DFT, it is intuitive to use their predictions as surrogate labels for configurations without DFT energies in order to resolve the label scarcity issue of training neural networks. However, there are two fundamental challenges. First, as

discussed in Sec. 2.2, while physics-based EIPs are designed to be accurate in specific regions of the configuration space and generalize better than those based on machine learning, they may still be inaccurate in other regions. Given an arbitrary configuration for which no DFT energy is available, it is unknown which EIP from a given set will yield the most accurate prediction and how large its error will be. Second, surrogating DFT energies with EIP energies inevitably introduces noise, and potentially outliers, into the training set that may have pathological effects.

**EIP prediction using an auxiliary classification model.**   In order to augment training sets with unlabeled configurations and their corresponding EIP-approximated energies, we use an auxiliary classification model to predict the best-performing EIP for a given configuration. For a configuration $C$, the classification model predicts a discrete distribution over the EIP set $\mathcal{P}' = \mathcal{P} \bigcup \{p_\emptyset\}$ that indicates their probability of being the most accurate EIP for $C$, i.e., $\mathbb{P}\left(\mathcal{P}' \mid C\right)$. We introduce a dummy EIP, $p_\emptyset$, to represent the case where none of the physics-based EIPs in $\mathcal{P}$ is predicted to approximate DFT to an accuracy level $c$, i.e., $\frac{1}{N_i}||E_i^p - E_i^{\text{DFT}}||_1 > c, \forall p \in \mathcal{P}$, where $N_i$ is the number of atoms in configuration $i$; throughout this work, $c$ is set to $0.1$. By excluding configurations that are labeled with the dummy class from the training set, $c$ acts as a confidence threshold to control the noise and outliers introduced by using the surrogate EIP energies.

The classification model consists of a representation learning module that converts an atomic configuration to fixed-length feature vectors (one for each atom in the configuration), a permutation-invariant readout function that aggregates them to form a feature vector describing the entire configuration, and a prediction module that maps the configuration representation to a set of probabilities. We train the classification model on $\mathcal{C}_{\text{DFT}}$ by optimizing a cross-entropy loss and apply it to $\mathcal{C}_{\text{EIP}}$. Configurations with a predicted EIP other than $p_\emptyset$ are assigned the corresponding EIP energy and are merged with the configurations that have DFT energies to arrive at the final training set. We denote the set of EIP-labeled configurations as $\hat{\mathcal{C}}_{\text{EIP}} = \{C_i, \{E_i^p\}_{p \in \mathcal{P}}, E_i^{\hat{p}_i}\}_{i=m+1}^{m+s}$ where $1 \leq s \leq n$ is the number of selected configurations not labelled by DFT, and $\hat{p}_i$ and $E_i^{\hat{p}_i}$ are the predicted best-performing EIP and its prediction on $C_i$.

**Regression with robust loss functions.**   We train the NN potential using configurations with ground-truth DFT energies and configurations with EIP energies selected by the classification model. In regression problems, models are usually trained by optimizing the mean square error (MSE) loss. The MSE loss is sensitive to outliers, as the magnitude of its gradient is linearly proportional to the difference between the predicted value and the ground truth value. To lessen the impact of outliers, we optimize the MSE loss on DFT-labeled configurations, while on EIP labeled-configurations, we optimize the Tukey biweight (bisquare) loss [6, 11, 5], i.e.,

$$\mathcal{L} = \frac{1}{m} \sum_{i=1}^{m} (E_i^{\text{DFT}} - \hat{E}_i)^2 + \frac{\alpha}{s} \sum_{i=m+1}^{m+s} l_{\text{Tukey}}(E_i^{\hat{p}_i} - \hat{E}_i) \tag{3}$$

where $\hat{E}_i$ is the model prediction of the energy for configuration $C_i$, and $\alpha > 0$ is a hyper-parameter that controls the contribution of EIP-labeled configurations to the loss and its gradient.

The Tukey biweight loss falls under the M-estimation method [20] and is intended to screen outliers using robust statistics of the regression residuals such as median absolute residuals (MAR) and suppress their influence on the gradient. The Tukey loss function is computed as

$$l_{\text{Tukey}}(r_i) = \begin{cases} \frac{k^2}{6}\left[1 - \left(1 - \left(\frac{r_i}{k}\right)^2\right)^3\right], & |r_i| \leq k \\ \frac{k^2}{6}, & |r_i| > k \end{cases} \tag{4}$$

where $r_i = E_i^{\hat{p}_i} - \hat{E}_i$ is the residual and $k$ is a tuning constant that is commonly set to $4.685\sigma$ to produce $95\%$ efficiency when the errors are normally distributed with standard deviation $\sigma$. To set the value of $k$, we estimate the standard deviation as $\hat{\sigma} = \text{MAR}/0.6745$. From Eq. 4, residuals with absolute values greater than $k$ are considered outliers and are rejected for gradient computation. In each gradient step, we sample from both DFT- and EIP-labeled configurations to form a batch for gradient back-propagation, and the MAR is estimated on residuals of both DFT- and EIP-labeled configurations in the batch. Since the MAR and $\hat{\sigma}$ are dynamically estimated during training, the Tukey loss does not introduce additional hyperparameters.

### 4.3 Multi-task Pretraining

We propose a multi-task pretraining strategy to encode the domain knowledge in physics-based EIPs to the parameters of the representation learning module by jointly predicting the set of EIP calculations for configurations with only EIP predictions, $\mathcal{C}_{\text{EIP}}$, and optimizing the following multi-task regression loss:

$$\mathcal{L} = \frac{1}{n|\mathcal{P}|} \sum_{p \in \mathcal{P}} \sum_{i=m+1}^{m+n} (\hat{E}_i^p - E_i^p)^2.$$

During pretraining, we couple the representation learning module with $|\mathcal{P}|$ prediction modules (MLPs) to generate predictions corresponding to different physics-based EIPs, i.e., $\hat{E}_i^p = f_{\text{pred}}^p \circ f_{\text{rep}}(C_i)$. The pretrained representation module is then fine-tuned with a randomly initialized prediction module for the downstream DFT prediction task. Note that the representation module could either be naively fine-tuned on configurations with DFT energies by optimizing an MSE loss, or on the training set generated by our proposed label augmentation method by optimizing Eq. 3.

Although transfer learning has been successful in various application domains, it could easily hinder model performance on the target task if the pretraining tasks are unrelated to the target task (negative transfer). We argue that predicting the output of physics-based EIPs is relevant and beneficial to the target task of predicting DFT energies. Although physics-based EIPs are not perfectly accurate across the space of all possible spatial arrangements of atoms, their functional forms incorporate prior physical information that allow them to correlate with DFT over this space. In Sec. 5, we empirically demonstrate that the multi-task pretraining strategy successfully encodes domain knowledge into configuration representations and creates a smoother DFT energy surface.

### 4.4 Combining Label Augmentation and Multi-task Pretraining

The label augmentation and multi-task pretraining methods outlined above can be combined with relative ease. The procedure is similar to the ordinary label augmentation strategy, but rather than using a randomly initialized representation module $f_{\text{rep}}$ for the final NN training, the representation module produced by the multi-task pretraining method is used during fine-tuning. Fig. 2 provides a schematic overview of both strategies, how they relate to one another, and how they can be combined.

## 5 Experiments

To test our methodology, we experiment with three datasets: the ANI-Al [32] dataset and the KIM-Si [22] dataset each with a single species, as well as a multispecies AgAu dataset [41]. Detailed descriptions of the three datasets are provided in Appendix A.2. For each dataset, we generate three splits by randomly assigning 20% of the DFT-labeled configurations as test sets and the other 80% as training sets. During training, we use 20% of the training set as a validation set for model selection. All of the reported experimental results are averaged over three different splits to avoid over-fitting to a specific split. We release our code [2] and the KIM-Si dataset [3] for reproducing our experimental results and continuous works.

### 5.1 Experimental Setting

#### 5.1.1 Neural Network-based Potentials

We evaluate our proposed strategies on two classes of neural network potentials that reflect the majority of machine learning potentials currently in use. The first represents atomic environments using pre-computed descriptors and learns non-linear transformations (MLPs) to map the descriptors to atomic embeddings, while the second uses GNNs to learn atomic representations from configuration graphs. We select one representative potential from each class. For our MLP-based potential, we use the Smooth Overlap of Atomic Positions (SOAP) [1] atomic environment descriptor together with a representation module and prediction module consisting of MLPs; we term this potential SOAPNet in later discussions. For our GNN-based potential, we select SchNet [30], CGCNN [46],

---

[2]`https://github.com/shuix007/EIP4NNPotentials`
[3]`https://doi.org/10.6084/m9.figshare.21266064`

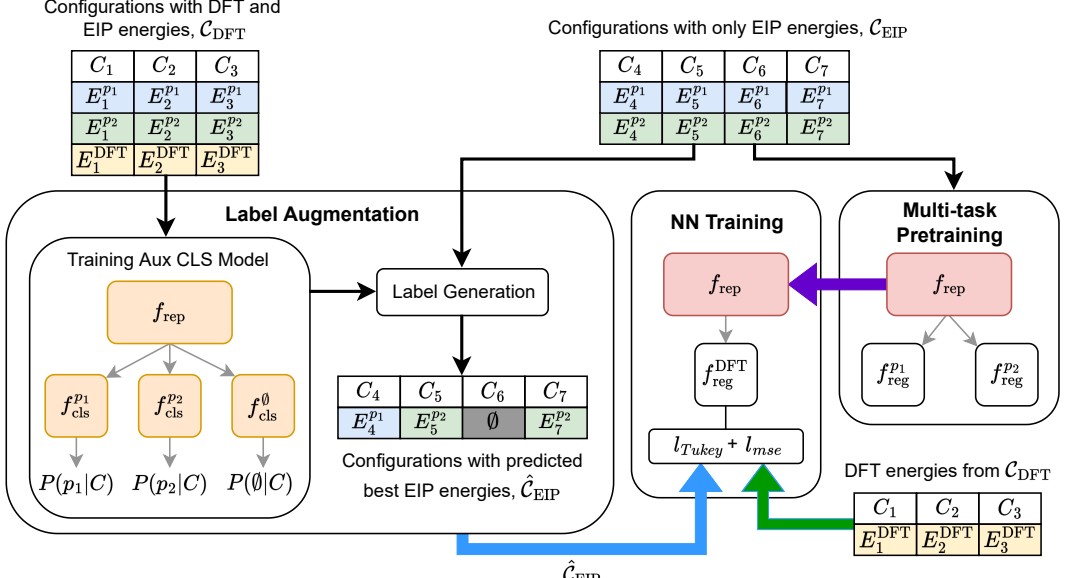

Figure 2: Illustration of the Label Augmentation (LA) and Multi-task Pretraining (MP) strategies and their usage in the training of NN-based potentials for the case of two physics-based EIPs, $p_1$ and $p_2$. In LA, a classifier is trained to predict the most accurate EIP energies $\hat{\mathcal{C}}_{\text{EIP}}$ for the unlabeled training instances, which are combined with the DFT energies from $\mathcal{C}_{\text{DFT}}$ in the loss function when training the final NN (blue arrow + green arrow). In MP, the representation module $f_{\text{rep}}$ is pretrained by simultaneously fitting the energies of each physics-based EIP before it is inherited as the initial state for the representation module in the final NN training, where the DFT-labeled instances are again used (purple arrow + green arrow). The two strategies can be combined by incorporating $\hat{\mathcal{C}}_{\text{EIP}}$ into the loss while also using multi-task pretraining to initialize $f_{\text{rep}}$ (blue arrow + purple arrow + green arrow).

and GemNet [14]. In our label augmentation experiments, we set the representation module of the auxiliary classification model to be the same kind as the corresponding NN potential, e.g., the classification model used for training the SchNet potential has a SchNet GNN as its representation module. Details about the model and training hyper-parameters can be found in Appendix A.4.

### 5.1.2 Selection of Physics-based EIPs

The physics-based EIPs used in our experiments were selected to encompass differing levels of functional complexity. Because physics-based EIPs are designed for specific elemental species (in this case, aluminum, silicon, and gold and silver systems), a different set of physics-based EIPs had to be chosen for each dataset. A total of ten physics-based EIPs were used for aluminum, eight for silicon, and two for the gold-silver system (see Tab. 2 in Appendix). They were taken mainly from the Open Knowledgebase of Interatomic Models (OpenKIM)[4] repository. [35, 36]. For detailed information, see Appendix A.3.

### 5.2 Experimental Results

### 5.2.1 Performance of Label Augmentation and Multi-task Pretraining

As shown in Tab. 1, our proposed strategies improve the performance of the four baseline NNs on the three benchmark datasets. In particular, the label augmentation strategy improves the baseline NNs by 5% to 51%, while the multi-task pretraining strategy improves the baselines by 2% to 55%. Combining the two strategies gives further improvement.

---

[4]`https://openkim.org/`

Table 1: Performance of the two proposed strategies on DFT energy prediction tasks. We report the configuration-level and atom-level mean absolute error (MAE, lower is better) in eV and eV/atom, respectively. We denote the label augmentation strategy by LA and the multi-task pretraining strategy by MP. Best performance is shown in bold. Cases where the training procedure failed due to running out of memory are marked OOM.

| | KIM-Si | | | ANI-Al | | | AgAu | | |
| | Config | Atom | Improv. | Config | Atom | Improv. | Config | Atom | Improv. |
|---|---|---|---|---|---|---|---|---|---|
| Best EIP | 1.6326 | 0.2524 | - | 46.4869 | 0.3561 | - | 4.4587 | 0.2063 | - |
| SOAPNet | 0.7706 | 0.0975 | - | 0.2153 | 0.0017 | - | 0.5422 | 0.0226 | - |
| +LA | 0.5595 | 0.0704 | 27.58% | 0.1786 | 0.0014 | 18.14% | 0.5067 | 0.0205 | 07.92% |
| +MP | 0.5717 | 0.0733 | 25.28% | 0.1744 | 0.0014 | 19.12% | 0.3962 | 0.0154 | 29.34% |
| +MP+LA | **0.5307** | **0.0657** | **31.88%** | **0.1697** | **0.0013** | **22.13%** | **0.3858** | **0.0154** | **30.23%** |
| SchNet | 0.4805 | 0.0718 | - | 0.1693 | 0.0014 | - | 0.7290 | 0.0290 | - |
| +LA | 0.4015 | 0.0549 | 19.99% | 0.0845 | 0.0007 | 51.24% | 0.6815 | 0.0266 | 07.33% |
| +MP | 0.4034 | 0.0569 | 18.40% | 0.1296 | 0.0010 | 26.00% | **0.3353** | **0.0130** | **54.65%** |
| +MP+LA | **0.3719** | **0.0490** | **27.17%** | **0.0816** | **0.0006** | **53.27%** | 0.3496 | 0.0135 | 52.80% |
| CGCNN | 0.9314 | 0.1410 | - | 0.2410 | 0.0019 | - | 1.6683 | 0.0625 | - |
| +LA | 0.7476 | 0.1050 | 22.61% | 0.1786 | 0.0014 | 25.44% | 1.6065 | 0.0589 | 04.71% |
| +MP | 0.8457 | 0.1253 | 10.16% | 0.2206 | 0.0017 | 07.80% | 1.4377 | 0.0532 | 14.38% |
| +MP+LA | **0.7435** | **0.1005** | **24.44%** | **0.1392** | **0.0011** | **41.65%** | **1.3857** | **0.0499** | **18.55%** |
| GemNet | 0.5138 | 0.0546 | - | OOM | OOM | OOM | 0.9257 | 0.0342 | - |
| +LA | 0.4691 | 0.0511 | 07.55% | OOM | OOM | OOM | 0.8381 | 0.0300 | 10.87% |
| +MP | 0.5024 | 0.0531 | 02.48% | OOM | OOM | OOM | **0.5057** | **0.0185** | **45.71%** |
| +MP+LA | **0.4651** | **0.0476** | **11.12%** | OOM | OOM | OOM | 0.6074 | 0.0218 | 35.30% |

### 5.2.2 EIP Energies as High-Quality Supervision Signals for Training NN Potentials

Recall that in the label augmentation strategy, an auxiliary classification model selects unlabeled configurations and predicts their corresponding best-performing physics-based EIPs, which are subsequently used to label them for the training of the NN potential. These configurations are then labeled by the predicted best-performing physics-based EIPs for NN potential training. We investigate the quality of EIP labels and the auxiliary classification model by training NN potentials on three augmented training sets in which the selected unlabeled configurations are labeled by three different sources: DFT-based energies, ground-truth best-performing-EIP energies, and predicted best-performing EIP energies. The DFT-labeled configurations (0.8K) are the same for the three training sets. We only conduct this experiment on the ANI-Al dataset, as all of its configurations have DFT energies available. The number of selected unlabeled configurations is shown in Tab. 6 (in Appendix).

Fig. 3a shows the performance of the NN potentials trained on the three augmented training sets and the original training set (where only DFT-labeled configurations are used). As shown in the figure, expanding the training set with ground truth DFT calculations (blue) greatly improves the baseline MAE (yellow bar, model trained on the original training set). Labeling configurations with the ground-truth best-performing physics-based EIPs (red bar) performs slightly worse than with DFT energies (blue bar) but still much better than the baseline (yellow bar). This demonstrates that physics-based EIPs are valuable sources of supervision signals for training NN-based potentials. Using the predicted best-performing physics-based EIPs for labeling (green bar) performs on par with the ground-truth best-performing physics-based EIP labeling (red), revealing the utility of the auxiliary classification model.

### 5.2.3 Importance of the Robust Tukey Loss

We next conduct experiments to investigate the importance of the Tukey loss and its ability to reject outliers during training. As before, we only conduct this experiment on the ANI-Al dataset, using DFT energies for unlabeled configurations to determine noise and outliers introduced by the predicted best-performing physics-based EIPs. We define an unlabeled configuration with a predicted best-performing physics-based EIP to be an outlier if the absolute difference between its physics-based

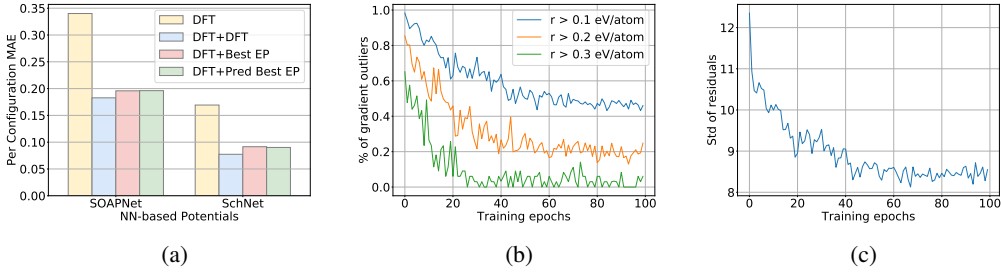

(a)                                (b)                                (c)

Figure 3: (a) Performance of NN potentials trained on the original training sets (yellow bar, DFT-labeled configurations only) and three augmented training sets whose unlabeled configurations are labeled by DFT energies (blue bar), ground-truth best-performing physics-based EIP energies (red bar), and predicted best-performing physics-based EIP energies (green bar). (b) Percentage of outliers used for computing gradient during training. (c) Standard deviation of residuals during training.

EIP energy and its DFT energy is larger than a threshold, i.e., $|r_i| = \frac{1}{N_i}|E_i^{\text{DFT}} - E_i^{\hat{p}_i}| > c$ where $N_i$ is the number of atoms in configuration $i$. We categorize the outliers as mild, normal, and severe by setting $c$ to $\{0.1, 0.2, 0.3\}$. The initial number of outliers introduced by the unlabeled configurations can be found in Tab. 6 (in Appendix). Figs. 3b and 3c show that the number of outliers of all kinds included for computing gradients decreases as the training proceeds, demonstrating that as the NN potential gets progressively more accurate, the Tukey loss can effectively eliminate outliers from the training set. We also conduct an ablation study by replacing the Tukey loss in Eq. 3 with the MSE loss. The results in Tab. 4 (in Appendix) show that the models' performance degrades without the Tukey loss.

### 5.2.4 Visualization of Pretrained Configuration Representations

Fig. 4 plots the t-SNE [38] 2D projections of the training silicon configuration representations colored by their per-atom DFT energies. The left-hand figure plots representations generated by a SchNet with random weights and the right-hand figure plots representations generated by a SchNet pretrained by the multi-task strategy. The representations generated by the randomly initialized SchNet do not exhibit any clear patterns and the energy surface is rough. In the right-hand figure, representations of the atomic cluster configurations (i.e., isolated groups of atoms) and the bulk configurations (crystals) are clearly separated and form clusters in the t-SNE 2D space. The per-atom DFT energy surface of the right-hand figure is much smoother, i.e., configurations with similar energies are close to one another after pretraining. This verifies our previous statement that, although physics-based EIPs lack complete generalizability, they nonetheless correlate reasonably well with DFT over atomic configuration space, and demonstrates that our proposed multi-task pretraining strategy successfully encodes domain knowledge into the NN-based potentials. T-SNE plots for SOAPNet on both datasets and SchNet on the ANI-Al dataset show similar patterns. These figures may be found in Appendix.

## 6 Conclusion

In this paper, we show that physics-based EIPs can be used to incorporate domain knowledge into machine learning EIPs by providing additional supervision signals on a large body of unlabeled training instances. Two generic strategies are formulated. The first, label augmentation, is based on weakly supervised learning and uses a classifier to select unlabeled training instances to supplement the original labeled training set. The second, multi-task pretraining, is based on transfer learning and uses the predictions of a set of physics-based EIPs on unlabeled training instances to pretrain a machine learning EIP that is subsequently fine-tuned on the original training set. Our methodology is proven using several experiments, including a cross-validation study that demonstrates significant performance gains for either strategy individually and further improvement when they are combined, a label augmentation study that shows that using EIP energies as surrogate labels provides nearly as much robustness as using additional DFT-labeled training instances, and a statistical experiment that illustrates the utility of the Tukey loss in diminishing noise and excluding outliers. Finally, we present

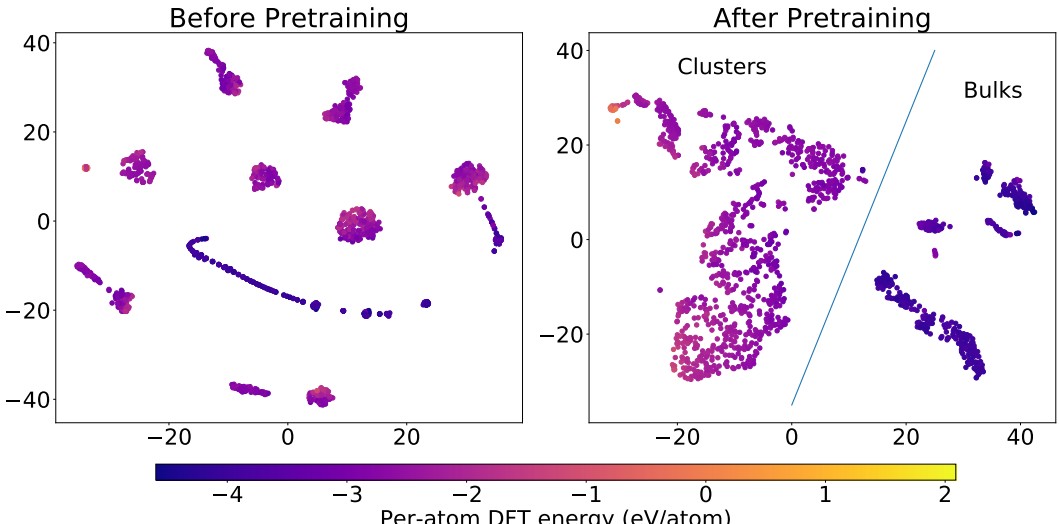

Figure 4: T-SNE plots of silicon configuration representations generated by a randomly initialized SchNet (left) and a SchNet pretrained with our proposed multi-task pretraining strategy (right). Configurations are colored by their per-atom DFT energies. Representations generated by the pretrained SchNet naturally form two clusters that correspond to the atomic cluster configurations (i.e., isolated groups of atoms) and the bulk configurations (i.e., crystals).

reduced-dimension visualizations that indicate the multi-task pretraining strategy yields training set feature vectors over which the energy varies smoothly.

One direction for future work is to replace the heuristic EIPs by more accurate but still computationally cheap methods, such as the semi-empirical tight binding method. These methods better approximate DFT and thus can lead to further improvement when combined with our proposed approaches.

# 7   Acknowledgement

This work was supported in part by NSF (1447788, 1704074, 1757916, 1834251, 1834332), Army Research Office (W911NF1810344), the startup funds from the Presidential Frontier Faculty Program at the University of Houston, Intel Corp, and Amazon Web Services. Access to research and computing facilities was provided by the Minnesota Supercomputing Institute. We thank the anonymous reviewers for their feedback during NeurIPS 2022 review process. We are grateful to Sijie He and Yingxue Zhou for their insightful discussion and inspiration.

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
