# Injecting Domain Knowledge from Empirical Interatomic Potentials to Neural Networks for Predicting Material Properties

**Zeren Shui**
University of Minnesota
shuix007@umn.edu

**Daniel S. Karls**[*]
University of Minnesota
karl0100@umn.edu

**Mingjian Wen**[*]
University of Houston
mjwen@uh.edu

**Ilia A. Nikiforov**
University of Minnesota
nikif002@umn.edu

**Ellad B. Tadmor**
University of Minnesota
tadmor@umn.edu

**George Karypis**
University of Minnesota / AWS
karypis@umn.edu

## A    Appendix

### A.1    Periodic Boundary Conditions

Under periodic boundary conditions (PBCs), the positions of atoms outside the simulation cell are obtained by generating periodic images of those within the cell through translations commensurate with its periodicity. This methodology is capable of modeling infinite systems because the interactions between atoms separated by more than a modest cutoff distance are very small and thus ignored when defining empirical models. This limited range of interaction gives rise to the concept of an *atomic environment*. The environment of a given atom consists of itself and all other atoms, including periodic images, that fall within a prescribed cutoff distance of it. The consequence of this locality is that an infinite system can be modeled exactly using a finite periodic cell so long as a sufficient number of periodic images surrounding it are explicitly accounted for. An example of PBCs for a two-dimensional square cell and a local atomic environment is illustrated in Fig. 5.

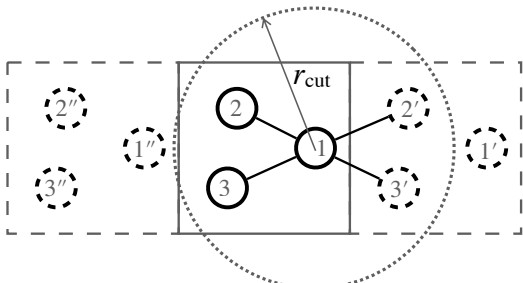

Figure 5: Illustration of periodic boundary conditions for a two-dimensional simulation cell (solid square) containing three atoms. For simplicity, only periodic images in the horizontal direction are shown. The local environment of atom 1 (dashed circle) contains all atoms and their periodic images that fall within a prescribed cutoff distance $r_{\text{cut}}$ of it.

---

[*]Equal contribution

36th Conference on Neural Information Processing Systems (NeurIPS 2022).

## A.2 Datasets

The ANI-Al dataset [8] consists of DFT energies of 6,352 configurations of aluminum in liquid, solid, and liquid-solid coexistence phases, each containing up to 250 atoms.

The KIM-Si dataset is a new dataset (soon to be published) that we generated for silicon comprising a total of 16,110 configurations. It builds upon the training set first used in [4] and contains a total of 14,510 perturbed bulk structures, 1,525 randomly generated atomic clusters ranging in size from two atoms to ten atoms, 3 ideal surface defects, and 72 nanostructures constructed by composing two-dimensional structures of silicon in the graphene and silicene geometries. In total, for the KIM-Si dataset, DFT energies are available for 510 of the bulk configurations and the 1600 non-bulk configurations.

The AgAu dataset [10] is developed to study the catalytic properties of AgAu binary nanoalloys. It consists of configurations for elemental Ag and Au, as well as AgAu binary alloy. The dataset is generated using an active learning strategy. First, an initial dataset of Ag, Au, and AgAu in body-centered cubic (BCC), face-centered cubic (FCC), and hexagonal close-packed (HCP) structures are labelled using DFT. Then, an ensemble of models are trained on the data and new configurations are selected to be further labelled by DFT based on the uncertainty obtained from the ensemble. This process is iterated multiple times. In this work, we randomly select 10k of the AuAg binary configurations to test our methodology on multispecies systems.

## A.3 Selection of Physics-based EIPs

We show the selected EIPs used in our experiments and their accuracy in Table 2 for reference. From the Table, we can observe that the physics-based EIPs used in label augmentation and multi-task pretraining perform significantly worse than the NN-based potentials and our proposed strategies. This is mainly because the functional form of an EIP is usually developed for specific configurations and cannot transfer very well to arbitrary structures. For example, the Tersoff (T3) potential employed on the KIM-Si dataset was developed for cubic diamond crystal structures and thus will not perform well for other structures in the test set.

Table 2: EIPs used in experiments and their accuracy. Standard deviations are shown in parentheses.

| Species | EIP name | Config | Atom | Link |
|---|---|---|---|---|
| Si | EDIP | 4.3063(0.1767) | 0.6071(0.0187) | https://doi.org/10.25950/545ca247 |
| | MEAM | 3.7315(0.0603) | 0.5235(0.0055) | https://doi.org/10.25950/b8dc8b23 |
| | SW (BalamaneHaliciogluTiller) | 21.3235(0.6292) | 2.7169(0.0765) | https://doi.org/10.25950/3dc2cb7f |
| | SW (ZhangXieHu) | 17.6129(0.4513) | 2.2711(0.0417) | https://doi.org/10.25950/32a4bf2c |
| | Tersoff (T2) | 1.6326(0.1262) | 0.2524(0.0177) | https://doi.org/10.25950/cadc4e78 |
| | Tersoff (T3) | 11.8825(0.3287) | 1.5889(0.0449) | https://doi.org/10.25950/d6e8a23e |
| | Tersoff (ErhartAlbe) | 6.4907(0.2466) | 0.8393(0.0279) | https://doi.org/10.25950/6aa22835 |
| | Tersoff (TMOD) | 5.4128(0.2036) | 0.7210(0.0259) | https://doi.org/10.1016/j.commatsci.2006.07.013 |
| Al | EMT | 50.8366(0.8829) | 0.3922(0.0053) | https://doi.org/10.25950/bdbaee6a |
| | Morse (LowCutoff) | 143.3241(1.4645) | 1.1433(0.0082) | https://doi.org/10.25950/977dc2ac |
| | Morse (MedCutoff) | 120.6183(1.4162) | 0.9711(0.0080) | https://doi.org/10.25950/474ccb33 |
| | Morse (HighCutoff) | 116.5143(1.4021) | 0.9400(0.0079) | https://doi.org/10.25950/45d9848f |
| | EAM (ErcolessiAdams) | 78.8916(1.7788) | 0.6105(0.0097) | https://doi.org/10.25950/bc2d2486 |
| | EAM (SturgeonLaird) | 79.4206(2.0645) | 0.6147(0.0115) | https://doi.org/10.25950/d62edb43 |
| | EAM (WineyKubotaGupta) | 95.8777(0.9002) | 0.7587(0.0054) | https://doi.org/10.25950/23542694 |
| | EAM (ZopeMishin) | 46.4869(0.6270) | 0.3561(0.0042) | https://doi.org/10.25950/26dbac6e |
| | EAM (Zhakhovsky) | 66.0950(1.3760) | 0.5091(0.0075) | https://doi.org/10.25950/c3a79c52 |
| | EAM (ZhouJohnsonWadley) | 69.1926(0.5550) | 0.5289(0.0020) | https://doi.org/10.25950/c775fc98 |
| AgAu | EAM (ZhouJohnsonWadley) | 5.7077(0.0942) | 0.2438(0.0077) | https://doi.org/10.25950/d77528cf |
| | EMT (JacobsenStoltzeNorskov) | 4.4587(0.1436) | 0.2063(0.0012) | https://doi.org/10.25950/485ab326 |

Most of the silicon physics-based EIPs were trained using atomic configurations similar to the bulk diamond configurations in our dataset. This raises the concern of information leakage, where the NN potentials indirectly learn information about the test set during training through supervision signals provided by the physics-based EIPs, since they were potentially trained on part of the test set. However, in the first two folds of our cross-validation, the perturbed diamond configurations constitute only 2.4% of the test set, and in the third fold constitute only 3.6%. In the case of aluminum, none of the physics-based EIPs used in our experiments were fitted to any configurations similar to those in the ANI-Al dataset. Altogether, we conclude that the impact of this effect is minimal.

Table 3: Performance of the two proposed strategies on DFT energy prediction tasks. We report the configuration-level and atom-level mean absolute error (MAE, lower is better) in eV and eV/atom, respectively. We denote the label augmentation strategy by LA and the multi-task pretraining strategy by MP. Standard deviations are shown in parentheses. Cases where the training procedure failed due to running out of memory are marked OOM.

| | KIM-Si | | ANI-Al | | AgAu | |
| | Config | Atom | Config | Atom | Config | Atom |
|---|---|---|---|---|---|---|
| SOAPNet | 0.7706 (0.0658) | 0.0975 (0.0012) | 0.2153 (0.0228) | 0.0017 (0.0002) | 0.5422 (0.0305) | 0.0226 (0.0016) |
| +LA | 0.5595 (0.0336) | 0.0704 (0.0026) | 0.1786 (0.0239) | 0.0014 (0.0002) | 0.5067 (0.0470) | 0.0205 (0.0021) |
| +MP | 0.5717 (0.0232) | 0.0733 (0.0038) | 0.1744 (0.0249) | 0.0014 (0.0002) | 0.3962 (0.0446) | 0.0154 (0.0019) |
| +MP+LA | 0.5307 (0.0375) | 0.0657 (0.0033) | 0.1697 (0.0193) | 0.0013 (0.0002) | 0.3858 (0.0398) | 0.0154 (0.0022) |
| SchNet | 0.4805 (0.0345) | 0.0718 (0.0096) | 0.1693 (0.0106) | 0.0014 (0.0001) | 0.7290 (0.0692) | 0.0290 (0.0026) |
| +LA | 0.4015 (0.0687) | 0.0549 (0.0042) | 0.0845 (0.0095) | 0.0007 (0.0001) | 0.6815 (0.0659) | 0.0266 (0.0026) |
| +MP | 0.4034 (0.0723) | 0.0569 (0.0115) | 0.1296 (0.0164) | 0.0010 (0.0001) | 0.3353 (0.0331) | 0.0130 (0.0012) |
| +MP+LA | 0.3719 (0.0706) | 0.0490 (0.0059) | 0.0816 (0.0064) | 0.0006 (0.0001) | 0.3496 (0.0295) | 0.0135 (0.0011) |
| CGCNN | 0.9314 (0.0379) | 0.1410 (0.0099) | 0.2410 (0.0318) | 0.0019 (0.0002) | 1.6683 (0.0851) | 0.0625 (0.0049) |
| +LA | 0.7476 (0.0516) | 0.1050 (0.0036) | 0.1786 (0.0241) | 0.0014 (0.0002) | 1.6065 (0.1314) | 0.0589 (0.0055) |
| +MP | 0.8457 (0.0611) | 0.1253 (0.0063) | 0.2206 (0.0380) | 0.0017 (0.0003) | 1.4377 (0.0852) | 0.0532 (0.0036) |
| +MP+LA | 0.7435 (0.0510) | 0.1005 (0.0043) | 0.1392 (0.0213) | 0.0011 (0.0002) | 1.3857 (0.1053) | 0.0499 (0.0044) |
| GemNet | 0.5138 (0.1097) | 0.0546 (0.0023) | OOM | OOM | 0.9257 (0.1133) | 0.0342 (0.0045) |
| +LA | 0.5024 (0.1225) | 0.0531 (0.0016) | OOM | OOM | 0.5057 (0.0691) | 0.0185 (0.0026) |
| +MP | 0.4691 (0.1296) | 0.0511 (0.0075) | OOM | OOM | 0.8381 (0.0567) | 0.0300 (0.0023) |
| +MP+LA | 0.4651 (0.1410) | 0.0476 (0.0041) | OOM | OOM | 0.6074 (0.0499) | 0.0218 (0.0022) |

## A.4 Experimental Settings

We set the number of hidden dimensions of all NNs to 128, the number of stacked NN layers in representation modules (GNNs and MLPs) to 5, and use the shifted softplus activation function in all nodes. We choose sum pooling to be the readout function and an MLP to be the prediction module. All of our models are optimized using the Adam algorithm [5] with a learning rate of 1e-3 and a batch size of 32. For the multi-task pretraining and the training of the auxiliary classification model, we use a slated triangular scheduler [3] for the initial warm up of the weights, and subsequently decrease the learning rate linearly. For training the NNs with DFT energies, we decrease the learning rate linearly from 1e-3 to 1e-5. All NNs are trained for 100 epochs. Results are reported on the checkpoint with the best validation performance. We use validation performance to select the hyperparameter $\alpha$ that controls the contribution of EIP-labeled configurations (cf. Section 4.2) from [0.01, 0.05, 0.1, 0.5, 1, 5, 10].

Our code for the (G)NN potentials and experiments is implemented using PyTorch [7]. The implementation of SchNet is modified from DGL [9], and DGL-LifeSci [6]. The implementation of CGCNN and GemNet are modified from PyG [2] and Open Catalyst Project [1] All experiments are conducted on a machine with an Intel(R) Core(TM) i9-10900F CPU and an Nvidia RTX-3090 GPU. Pretraining SchNet and CGCNN on the ANI-Al and the KIM-Si dataset takes 1.5hr and 0.5hr, respectively, while retraining SOAPNet on both datasets takes a few minutes. The time cost of label augmentation depends largely on the number of unlabeled configurations added to the training set. In our case, experiments in all settings take less than 2hrs to finish.

## A.5 Additional Experimental Results

### A.5.1 Performance of the Two Strategies with Standard Deviation

We report performance of the two strategies on an average over three runs on three random splits and report the mean/std in Tab. 3

### A.5.2 Ablation Study of the Tukey Loss

Tab. 4 shows the configuration-level MAE for the KIM-Si and ANI-Al datasets with and without using the Tukey loss.The results show that the models' performance degrades without the Tukey loss.

Table 4: Configuration-level MAE (eV) with and without the Tukey loss.

| | KIM-Si | | ANI-Al | |
| | SOAPNet | SchNet | SOAPNet | SchNet |
|---|---|---|---|---|
| w/o Tukey | 0.5556 | 0.4374 | 0.1906 | 0.1031 |
| w/ Tukey | 0.5595 | 0.4015 | 0.1786 | 0.0845 |

Table 5: MAE (eV and eV/atom) on training sets expanded by configurations with different confidence level.

| | | KIM-Si | | ANI-Al | |
| | Confidence | Config | Atom | Config | Atom |
|---|---|---|---|---|---|
| SOAPNet | Low | 0.5793 | 0.0732 | 0.2174 | 0.0017 |
| | Medium | 0.5728 | 0.0725 | 0.2133 | 0.0017 |
| | High | 0.6196 | 0.0778 | 0.2361 | 0.0019 |
| SchNet | Low | 0.4817 | 0.0657 | 0.1140 | 0.0009 |
| | Medium | 0.4916 | 0.0668 | 0.1130 | 0.0009 |
| | High | 0.5004 | 0.0740 | 0.1242 | 0.0010 |

### A.5.3 Effect of Expanding Training Set with Different Confidence Level

We expand the training set with DFT energies with EIP-labeled configurations with different confidence level computed by the auxiliary classification model, i.e., $P(\hat{p} \mid C)$ (see Section 4.2). We first sort the configurations by their confidence scores and assign them to different confidence groups. Configurations with a confidence score higher than the 0.66 quantile are assigned to high confidence group, configurations with a confidence score lower than the 0.33 quantile are assigned to low confidence group, other configurations are assigned to the medium confidence group. We expand the DFT-labeled configurations to three different training sets by adding configurations belonging to different groups. Results are shown in Tab. 5. The results suggest that the unlabeled configurations with medium confidence contribute the most to the training while configurations with high confidence contribute the least. This is because the high confidence configurations may be very similar to the configurations in the training set and thus do not provide extra information. Low and medium confidence configurations are more dissimilar than those in the original training set and can provide more information.

### A.5.4 Number of Outliers and Configurations Selected By the Classification Model

We show the number of configurations and outliers introduced by the auxiliary classification model in Tab. 6. Results in the table show that the auxiliary classification model reaches an accuracy of 75% to 78% for screening configurations and selecting reasonable EIPs for labeling them. This demonstrates the utility of the classification model.

### A.5.5 Influence of the Selected Potentials

We conducted an ablation study on the KIM-Si dataset to investigate the influence of the selected set of EIPs on our two proposed strategies. Among the eight EIPs for Si, we selected two EIPs that work the best and the worst on the KIM-Si dataset, respectively. We apply LA and MP with the two EIPs separately and jointly (for a total of three experiments) on SOAPNet and SchNet and report

Table 6: Average number of configurations and outliers selected by the classification models on the ANI-Al dataset.

| | #Mild | #Normal | #Severe | #Selected | #Unlabeled |
|---|---|---|---|---|---|
| SOAPNet | 226 | 42 | 5 | 1211 | 5081 |
| SchNet | 271 | 50 | 9 | 1300 | 5081 |

Table 7: MAE (eV and eV/atom) of different selected set of EIPs on Si. Numbers in the brackets indicate the number of EIPs used for the experiments.

|  |  | SOAPNet | | SchNet | |
|---|---|---|---|---|---|
|  |  | Config | Atom | Config | Atom |
|  | Baseline | 0.7582 | 0.0987 | 0.5021 | 0.0745 |
| +LA | Default (8) | 0.5673 | 0.0725 | 0.4114 | 0.0545 |
|  | Best EIP (1) | 0.6084 | 0.0797 | 0.4395 | 0.0614 |
|  | Worst EIP (1) | 0.766 | 0.0995 | 0.5245 | 0.0782 |
|  | Mix (2) | 0.5878 | 0.0767 | 0.4286 | 0.0596 |
| +MP | Default (8) | 0.5679 | 0.0741 | 0.4217 | 0.0545 |
|  | Best EIP (1) | 0.6897 | 0.0892 | 0.4127 | 0.0561 |
|  | Worst EIP (1) | 0.6734 | 0.0881 | 0.4116 | 0.0559 |
|  | Mix (2) | 0.6549 | 0.0856 | 0.3659 | 0.0476 |

their results in Tab. 7. From the table we can observe that, label augmentation (LA) with a good EIP improves the baseline performance while LA with a bad EIP does not hurt the performance very much thanks to the auxiliary classification model and the robust Tukey loss. When applying LA on a mixture of good and bad EIPs, our strategy is able to select good from bad and provide performance boost. Moreover, the more (good) EIPs leveraged in LA, the better performance it can give. For multi-task pretraining, we observe that the number and quality of the EIPs do not influence the performance by too much.

### A.5.6 Visualization of Pretrained Configuration Representations

We show the t-SNE 2D projection plots for all datasets and NN potentials in Figs. 6, 7, and 8. These plots show that the multi-task pretraining successfully injects domain knowledge into the representation module of NN potentials and creates a smoother DFT energy surface.

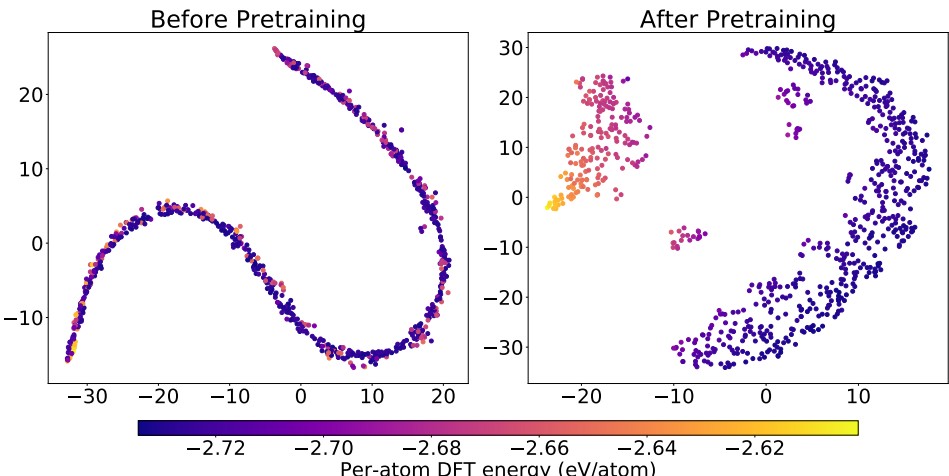

Figure 6: T-SNE plots of aluminum configuration representations generated by a randomly initialized SchNet (left) and a SchNet pretrained with our proposed multi-task pretraining strategy (right).