# OpenReview forum: "Injecting Domain Knowledge from Empirical Interatomic Potentials to Neural Networks for Predicting Material Properties"
_NeurIPS.cc/2022/Conference — NeurIPS 2022 Accept_

### Official Review · Reviewer_DKkp · 2022-07-04

**Rating:** 6
**Confidence:** 4
**Soundness:** 3 good
**Presentation:** 4 excellent
**Contribution:** 3 good

**Summary:**

This paper proposes to improve the prediction performance of the expensively computed material energy from density functional theory (DFT) by making use of the cheaply computed material energy from empirical interatomic potentials (EIP). Two strategies are proposed to use EIP based labels, including label augmentation and multi-task pretraining. Experimental results show that these two strategies can both improve the prediction performance of two models.

**Questions:**

(1) In label augmentation strategy, what is the prediction accuracy of the auxiliary classification model? Since the auxiliary classification model will determine which EIP-approximated energies will be treated as DFT labeled energies, it is important for it to have a high accuracy.
(2) In experiments, how label augmentation and multi-task pretraining strategies are combined? Are "+MP+LA" results obtained by training two models following the two strategies and ensembling their results, or by some other combining methods?

**Strengths And Weaknesses:**

Strengths:
(+) This work gives a very meaningful exploration of improving the neural network model for DFT computation prediction with large amounts of EIP labeled data given that DFT labeled data is limited. The success of the proposed method can motivate researchers working on developing machine learning models for DFT prediction to consider using cheap data source to improve their models.
(+) The writing and organization of the paper is clear and easy to follow.

Weaknesses:
(-) In experiments, SOAPNet and SchNet are used as prediction models. However, they are designed for molecules, not the best models for material data. It is better to use existing material property prediction models, such as CGCNN [1], in experiments.
(-) Necessary description about how label augmentation and multi-task pretraining strategies are combined in experiments is lacking.


[1] Xie, Tian, and Jeffrey C. Grossman. "Crystal graph convolutional neural networks for an accurate and interpretable prediction of material properties." Physical review letters 120.14 (2018): 145301.

---

> ### Author Response · Authors · 2022-08-02
> **Response to reviewer DKkp**
>
> We want to thank the reviewer for carefully reviewing our manuscript, providing insightful feedback and suggestions that help us to improve our work. We have leveraged the suggestions in our revised manuscript.
>
> In summary, the reviewer has three concerns/questions/suggestions in regard to the manuscript: 1) The selected baselines (SOAPNet and SchNet) were originally developed for molecular property prediction in contrast to models that are developed for material property prediction (e.g., CGCNN); 2) How are the two proposed strategies of label augmentation and multitask pretraining combined?; and 3) What is the accuracy of the auxiliary classification model?
>
> We hereby answer the reviewer’s questions:
>
> 1) We added CGCNN as a baseline and observed similar results to SOAPNet and SchNet.  Results for CGCNN are provided in Table 1 of the revised manuscript. We also note that the SOAP descriptor framework has been successfully applied to a range of material types including crystals, amorphous bulks, and liquids, cf. https://doi.org/10.1103/PhysRevX.8.041048.
>
> 2) We have added a new Figure 2 and a new Section 4.4  to the revised manuscript to make this more clear. In summary, in label augmentation (LA), we first train the auxiliary classification model (randomly initialized) and select the best performing EIPs. Using the selected EIPs, we label extra configurations without DFT energies. These configurations labeled by the selected EIPs and the configurations with DFT energies are then used to directly train a randomly initialized neural network potential.
>
>     In multitask pretraining (MP), we first apply the multi-task pre-training strategy to pre-train a randomly initialized neural network potential using only EIP- labeled configurations. Note, contrary to LA where only the selected EIPs are used to provide extra labels, all available EIPs are used to provide labels to pretrain the neural network potential. Then, we finetune the pre-trained model using only the DFT- labeled configurations.
>
>     LA+MP combines the two. We first pretrain the model in exactly the same way as in MP. However, in the finetuning stage, in addition to the DFT-labeled configurations, we also use the configurations labeled by the best-performing EIPs (determined via the auxiliary classification model) inside of a Tukey loss function contribution, as in LA.
>
> 3) We agree that the accuracy of the auxiliary classification model is very important in our method. The accuracy of the classification model to find the most accurate physics-based EIPs is around 75% to 78%. In Appendix A.5.4 and Table 5, we show the number of configurations selected by the auxiliary classification model and the number of outliers introduced. In addition, the Tukey loss can effectively filter out the inaccurate EIP labels and then exclude them during training as demonstrated in Figure 3(b).
>
> We hope our response and revision answer your question and resolve your concern about our submission.

---

> > ### Comment · Reviewer_DKkp · 2022-08-04
> > **Response to Authors**
> >
> > Dear authors,
> >
> > Thanks for your responses here. Your responses have addressed my concerns well so I remain in favor of accepting the paper.

---

### Official Review · Reviewer_29j7 · 2022-07-07

**Rating:** 5
**Confidence:** 3
**Soundness:** 2 fair
**Presentation:** 3 good
**Contribution:** 2 fair

**Summary:**

In this manuscript, the authors proposed to incorporate domain knowledge into machine learning empirical interatomic potentials with two techniques, a weakly supervised learning based on auxiliary classifiers and a pretraining/fine-tuning mode based on transfer learning. Their experiment results have shown a comprehensive outperformance over the baseline methods on systems with a single atomic species.

**Questions:**

1. Please use some of the SOTA neural architectures as the base models.
2. Multiple species systems have been included in ANI and some new datasets such as OC20, and methods such as SchNet, DimNet, ForceNet, etc. have been evaluated on these systems. Why do the authors need to leave this as a future direction?


**Ethics Review Area:**

["I don’t know"]

**Limitations:**

I did not find any potential negative societal impact.

**Strengths And Weaknesses:**

On the strengths side, the presented method attempts to solve an import problem by leveraging unlabeled training instances generated from EIPs. The paper is well written and easy to follow. The main ideas are clearly explained and the empirical evaluation protocols and results are well presented.

However, both technical and theoretical contributions are inadequate. The auxiliary task modules and pre-training strategies have been widely applied to the GNNs. Albeit effective, these techniques are quite straightforward and lack technical motivation and insights.
Furthermore, the evaluation of the experiment is weak. The authors chose two old methods (SOAPNet in 2013 and SchNet in 2017) for validation and cannot show that these strategies are still valid in the current SOTA frameworks (e.g., DimNet++, arXiv:2003.03123 and GemNet arXiv:2106.08903). The evaluation conducted only on the single atom system also raises serious concerns about its generalization ability.

Overall, I think this work may be more appropriate for a journal in computational chemistry rather than a machine learning conference.

---

> ### Author Response · Authors · 2022-08-02
> **Response to reviewer 29j7**
>
> We want to thank the reviewer for carefully reviewing our manuscript, providing insightful feedback and suggestions that help us to improve our work. We have leveraged the suggestions in our revised manuscript.
>
> In summary, the reviewer has three major concerns/questions/suggestions in regard to the manuscript: 1) The proposed strategies lack technical motivation, insights, novelty, and theoretical contributions; 2) Baselines used in the experiments are not state-of-the-art; 3) Datasets used in the experiments are limited to single species systems.
>
> We hereby answer the reviewer’s questions:
>
> 1) We respectfully disagree with the reviewer's opinion that our proposed strategies lack technical motivation and novelty. As mentioned by the reviewer, we are trying to solve the label scarcity problem in the specific context of training NN-based empirical potentials, which is relevant to both the materials science and machine learning communities. For this problem, a clear motivation exists to use EIPs to cheaply label training data because they are designed with the intent of approximating the DFT energy surface in at least certain regimes of configuration space. As our results illustrate, predicting EIP energies is a relevant pretraining task, preventing the issue of negative transfer encountered by previous authors [1]. With regard to our contribution to the machine learning community, the implementation proposed in our label augmentation strategy, where a classification subproblem is solved and outliers are filtered out, can be applied when using any arbitrary collection of additional models (of varying accuracy across the domain of interest) to supplement the training set with cheaply labeled data. Altogether, our proposition of using EIPs as surrogate DFT supervision signals is effective, efficient, and readily usable by any NN-based potential, and has not been explored until the present work. We believe this constitutes a contribution significant enough to warrant publication in this venue.
>
> [1] Hu, Weihua, Bowen Liu, Joseph Gomes, Marinka Zitnik, Percy Liang, Vijay Pande, and Jure Leskovec. "Strategies for pre-training graph neural networks." arXiv preprint arXiv:1905.12265 (2019).
>
> 2) We added two additional baseline models, CGCNN and GemNet to our experiments, as suggested by the reviewer. We observe improvements of our proposed strategies over the two new baselines. The experimental results are updated in Table 1.
>
> 3) Thanks for the suggestion! We’ve added a new dataset from a multi-species system: Au and Ag. The results are aligned with those with the Si and Al dataset: label augmentation (LA) and multitask pretraining (MP) can both significantly improve model performance. Detailed new results are included in Table 1.
>
> We hope our response and revision answer your question and resolve your concerns about our submission.

---

> > ### Comment · Reviewer_29j7 · 2022-08-05
> > **Response to authors**
> >
> > Dear authors,
> >
> > Thanks for your response and great efforts in the experiments. I think the authors have addressed most of my concerns, i.e., demonstrating that the modules they added are useful for the SOTA GNN models and showing that the model can be extended for multiple species systems.  Considering that the author has solved two technical problems I had, I will raise the score to borderline acceptance. However, I still maintain that its technical tastes are better suited to a journal of computational chemistry.

---

### Official Review · Reviewer_ZTGs · 2022-07-08

**Rating:** 6
**Confidence:** 3
**Soundness:** 3 good
**Presentation:** 3 good
**Contribution:** 3 good

**Summary:**

This paper propose to "inject" the domain knowledge in empirical interatomic potentials (EIPs) into neural networks by using the data generated by EIPs. EIPs is much faster than DFT, and reasonably accurate. However,
* multiple EIPs may be applicable (which EIP to trust?)
* their accuracy varies in the configuration space (when should we trust EIPs?).

Two strategies are presented:

**LA**: label augmentation (semi-supervised learning).

A classifier is trained to jointly handle the two issues above.
* it predicts the best-performing EIP for a given configuration. (which EIP to trust?)
* If none is sufficiently accurate, outputs a dummy indicator. (when should we trust EIPs?).

This classifier is trained on configurations where DFT data is available (how much can it be generalized to unseen configurations?). This classifier is then used to augment data using the predicted best-performing EIP on each configuration. This builds an augmented dataset consisting of both EIP samples and DFT samples. As EIP labels are less accurate, a Tukey loss is used as it's less sensitive to outliers (as it's capped). MSE is used if the label comes from DFT.

**MP**: multi-task pretraining (transfer learning)

instead of only using the label from the predicted best-performing EIP, this strategy use all EIPs labels in pretraining in a multi-task way. Then DFT data is used to finetune the model.

The strategies are then demonstrated with two typical model backbones (SOAPNet as descriptors + MLP, and SchNet as xyz + GNN, ) on two material datasets (KIM-Si for silicon, and ANI-Al for aluminum). Both datasets are energy prediction tasks. Both strategies are shown to be able to reduce MAE and combining MP and LA can achieve even more improvement.

The main contribution is the two strategies proposed and demonstrated to help neural networks with EIPs data, which is usually much cheaper than DFT.


**Questions:**

* how exactly are MP and LA combined? firstly train on MP and then train with LA?
* how will performance change if a different/smaller set of EIPs are used.
* is the learned model more accurate than EIPs? (e.g. for configurations with DFT labels)

**Limitations:**

* see weakness. The sensitivity of the performance to the selected set of EIPs is not clearly addressed.


**Strengths And Weaknesses:**

**Strengths**:

* They propose two strategies shown to improve neural networks with EIPs data, which is usually much cheaper than DFT.
* They demonstration with both "descriptors + MLP" and "xyz + GNN" models. These two cases are very representative and therefore support the significance of their results.
* Several designs are reasonable: (e.g., dummy EIP, Tukey loss)

**Weaknesses**:

* The sensitivity of the performance to the selected set of EIPs (8 for silicon and 10 for aluminum as in the paper) is not clearly addressed. How will the performance change if only 2 or 3 EIPs are selected? What's the trend of model performance as the number of EIPs increases?
* can be more sound if the author compare the performance between the learned models vs. the EIPs used to build dataset. This can be evaluated for configurations with DFT labels.

some typos:
* line15, "bolster" => "booster"
* line165, "DFT and EIP energies" => "DFT energies"

---

> ### Author Response · Authors · 2022-08-02
> **Response to reviewer ZTGs**
>
> We want to thank the reviewer for carefully reviewing our manuscript, providing insightful feedback and suggestions that help us to improve our work. We have leveraged the suggestions in our revised manuscript.
>
> In summary, the reviewer has four major concerns/questions/suggestions in regard to the manuscript: 1) Sensitivity of the performance of the proposed strategies with respect to the selection of EIPs; 2) Comparison of the accuracy between the proposed methods with the EIPs used; 3) How are LA and MP combined with each other?; and 4) Typos
>
> 1) Thanks for raising this interesting question. We conducted an ablation study on the KIM-Si dataset to investigate the influence of the selected set of EIPs, and believe this helps improve our work. Among the eight EIPs for Si, we selected two EIPs that work the best and the worst on the KIM-Si dataset, respectively. We apply LA and MP with the two EIPs separately and jointly (for a total of three experiments) on SOAPNet and SchNet. We have updated our manuscript and added the results in Table 6. As the table shows, label augmentation (LA) with a good EIP improves the baseline performance while LA with a bad EIP does not hurt the performance very much thanks to the auxiliary classification model and the robust Tukey loss. When applying LA on a mixture of good and bad EIPs, our strategy is able to select good from bad and provide a performance boost. Moreover, the more (good) EIPs leveraged in LA, the better performance it can give. For multi-task pretraining, we observe that the number and quality of the EIPs do not influence the performance by too much. We believe that this is because a bad EIP can still provide a label that falls into a reasonable range and inject knowledge to the NN-based potentials by warming up their parameters.
>
> 2) We have revised our manuscript by adding the prediction accuracy of the EIPs used in Tables 6, 7, and 8 in the appendix. We have also added the best-performing EIPs of each dataset in Table 1 for comparison with the NN-based EIPs and our proposed strategies. From the Tables, we can observe that the physics-based EIPs used in label augmentation and multi-task pretraining perform significantly worse than the NN-based potentials and our proposed strategies. This is mainly because the functional form of an EIP is usually developed for specific configurations  and cannot transfer very well to arbitrary structures. For example, the Tersoff potential employed on the KIM-Si dataset was developed for cubic diamond crystal structures and thus it won’t do a good job for other structures that are in the test set.
>
> 3) We’ve added a new Figure 2 and a new Section 4.4  to the revised manuscript to make this more clear. In summary, in label augmentation (LA), we first train the auxiliary classification model (randomly initialized) and select the best performing EIPs. Using the selected EIPs, we label extra configurations without DFT energies. These configurations labeled by the selected EIPs and the configurations with DFT energies are then used to directly train a randomly initialized neural network potential.
>
>     In multitask pretraining (MP), we first apply the multi-task pre-training strategy to pre-train a randomly initialized neural network potential using only EIP-labeled configurations. Note, contrary to LA where only the selected EIPs are used to provide extra labels, all available EIPs are used to to provide labels to pretrain the neural network potential. Then, we finetune the pre-trained model using only the DFT-labeled configurations.
>
>     LA+MP combines the two. We first pretrain the model in exactly the same way as in MP. However, in the finetuning stage, in addition to the DFT-labeled configurations, we also use the configurations labeled by the best-performing EIPs (determined via the auxiliary classification model) inside of a Tukey loss function contribution, as in LA.
>
> 4) We have changed the word “bolster” in the introduction section to “increase.” The second bit of text pointed out on line 165 is actually not a typo. The set C_DFT consists of a total of `m` atomic configurations for which both DFT and EIP energies are available. The fact that EIP energies are available for these configurations is critical because it is this set of configurations that are used to train the auxiliary classifier in our label augmentation method. We considered using a different variable name such as “C_{DFT+EIP}” but thought this hurt overall readability. Because the EIP energies are extremely cheap to compute, we primarily wanted to emphasize the fact that DFT energies are available for these configurations.
>
> We hope our response and revision answer your question and resolve your concern about our submission.

---

### Official Review · Reviewer_znL5 · 2022-07-11

**Rating:** 7
**Confidence:** 4
**Soundness:** 3 good
**Presentation:** 3 good
**Contribution:** 3 good

**Summary:**

The authors introduce two strategies for predicting interatomic potentials. The first one is based on label augmentation. In this case, an auxiliary training is performed to classify the best-performing physics-based empirical interatomic potentials (EIP) for a given atomic configuration. If a given configuration (C) results in an a reliable energy E (there is also a label for no good classification), then the pair C-E is appended to the DFT-E training set. This strategy yields a performance boost from 18% to 51 %. The second method is based on transfer learning, where a NN is trained using EIP alone and then fine-tuned based on DFT energies. In this case, the improvements are from 18% to 26%.


**Questions:**

1. Is the proposed method potentially applicable to NequIP? If so, what would the expected outcome be?

2. The paper would be stronger if at least another dataset is considered.

3. It is not always clear if EIP is machine-learned or physics based. I suggest using two different nomenclatures for the two cases.

4. A figure in the main text would help understand the paper. The figure should show how the two NNs are connected to each other (e.g. for the first approach through label augmentation)




**Limitations:**

The authors state that their work is limited to single-specie materials (Si and Al in this case) and that future work will possible include multi-species ones. I believe limitations are adequately addressed.

**Strengths And Weaknesses:**

Strengths:

The paper proposes a neural network potential based on physics-based EIP and DFT-energy labelled dataset. The novelty of the manuscript is exploiting this multifidelity data using label augmentation, a novel approach for neural-network potentials. Also, the increase in the performance is obtained with little cost in the computational load (mainly the auxiliary training).

Weaknesses:

1. There is no comment on how this potential generalizes outside the selected species.

2. The methodology is limited to only two single-specie materials.

2. The paper is overall well-presented, but it lacks clarity is several aspects (see below for more details).

---

> ### Author Response · Authors · 2022-08-02
> **Response to reviewer znL5**
>
> We want to thank the reviewer for carefully reviewing our manuscript, providing insightful feedback and suggestions that help us to improve our work. We have leveraged the suggestions in our revised manuscript.
>
> In summary, the reviewer has four major concerns/questions/suggestions in regard to the manuscript: 1) How do the developed potentials generalize to other species? 2) Only use single species datasets in experiments; 3) How does the proposed method work on NequIP? 4) Confusing terminology and illustration suggestions.
>
> We hereby answer the reviewers’ questions:
> 1) A potential that is trained on one set of species is not expected to work well on another set of species.
> 2) Thanks for the suggestion! We’ve added a new dataset with two species: Au and Ag. The results are aligned with those with the Si and Al dataset: label augmentation (LA) and multitask pretraining (MP) can both significantly improve model performance. Detailed new results are included in Table 1.
> 3) Yes, the label augmentation (LA) and multitask pretraining (MP) methods proposed in this work can be applied to train any neural network-based potential, regardless of whether it uses invariant or equivariant features. To demonstrate their effectiveness on equivariant models, we’ve applied the strategies to NequIP and found that both LA and MP are effective. Specifically, using LA improves performance by 4% to 27% and using MP improves performance by up to 27%, as shown in the preliminary results below.
>
> |        |          | KIM-Si     |          | AgAu          |          |
> | ------ | -------- | ---------- | -------- | ------------- | -------- |
> |        |          | Per config | Per atom | Per config    | Per atom |
> | NequIP | Baseline | 0.7857     | 0.0962   | 2.2555        | 0.0892   |
> |        |   MP     | 0.7549     | 0.0955   | 1.6393        | 0.0662   |
> |        |   LA     | 0.5691     | 0.0716   | 2.1754        | 0.0843   |
> |        |   MP+LA  | 0.6416     | 0.0758   | 1.5863        | 0.0613   |
>
> (Due to time constraints, we are unable to repeat this experiment for different datasets and cross-validation splits, and so we have not incorporated these results into the paper). In addition to NequIP, we also experimented on two more recent models: CGCNN and GemNet. Again, the results are similar to previous ones. Detailed new results are included in Table 1.
>
> 4) As discussed in Section II, we use the term “physics-based potentials” to refer to traditional EIPs with analytic functional forms that bear some physical knowledge/constraints, and the term “machine learning potentials” (and, more specifically, “neural network potentials”) to refer to more recent potentials based on machine learning approaches.
> However, as the reviewer points out, in some places of the manuscript, especially sections 4 and 5, the term “EIP” is used to refer to physics-based potentials and this can lead to confusion. We made several edits in the manuscript to avoid such confusion:
> - Title of section 2.2 changed from “Physics-based Empirical Interatomic Potentials” to “Physics-based Potentials”;
> - Title of section 4.3 changed from “Multi-task Pretrain with Empirical Potential Predictions” to “Multi-task Pretraining”;
> - In section 4, “Our goal is to use this data to train an NN regression potential…” changed to “Our goal is to use this data from physics-based EIPs to train an NN-based EIP…”.
> - In addition, in multiple places in sections 4 and 5, the use of “EIPs” is changed to “physics-based EIPs” to make it more clear.
>
> Moreover, we want to thank the reviewer for the suggestion of an illustration figure. We agree that a figure would be very helpful for readers to understand our methods. We added Figure 2 in the revised manuscript to illustrate the label augmentation and  multi-task pretraining strategies, how they relate to one another, and how they can be combined. We hope our response and revision answer your question and resolve your concern about our submission.

---

> > ### Comment · Reviewer_znL5 · 2022-08-08
> > **Response to authors**
> >
> > Dear Authors,
> >
> > Thanks for successfully addressing my concerns. I will raise my rating to 7.

---

### Meta-Review · Area_Chair_MoAL · 2022-08-26

**Recommendation:** Accept
**Confidence:** Certain

**Metareview:**

This paper proposes two strategies for injecting domain knowledge into neural networks for predicting material propreties, these strategies lead to substantial accuracy gains. All reviewers had positive feedback on the paper and their suggestions helped improving the paper and the experiments. Accept

**Award:**

No

---

### Decision · Program_Chairs · 2022-09-14

Accept